# Context-Aware Lossless and Lossy Compression of Radio Frequency Signals

**DOI:** 10.3390/s23073552

**Published:** 2023-03-28

**Authors:** Aniol Martí, Jordi Portell, Jaume Riba, Orestes Mas

**Affiliations:** 1Departament de Teoria del Senyal i Comunicacions, Universitat Politècnica de Catalunya (UPC), Jordi Girona 1-3, 08034 Barcelona, Spain; aniol.marti@upc.edu (A.M.); jaume.riba@upc.edu (J.R.); orestes.mas@upc.edu (O.M.); 2Institut de Ciències del Cosmos (ICCUB), Universitat de Barcelona (IEEC-UB), Martí i Franquès 1, 08028 Barcelona, Spain; 3DAPCOM Data Services, Vilabella Centre de Negocis, Vilabella 5-7, 08500 Vic, Spain

**Keywords:** data compression, radio frequency compression, spectral estimation, software-defined radio (SDR), spectrum sensing

## Abstract

We propose an algorithm based on linear prediction that can perform both the lossless and near-lossless compression of RF signals. The proposed algorithm is coupled with two signal detection methods to determine the presence of relevant signals and apply varying levels of loss as needed. The first method uses spectrum sensing techniques, while the second one takes advantage of the error computed in each iteration of the Levinson–Durbin algorithm. These algorithms have been integrated as a new pre-processing stage into FAPEC, a data compressor first designed for space missions. We test the lossless algorithm using two different datasets. The first one was obtained from OPS-SAT, an ESA CubeSat, while the second one was obtained using a SDRplay RSPdx in Barcelona, Spain. The results show that our approach achieves compression ratios that are 23% better than *gzip* (on average) and very similar to those of FLAC, but at higher speeds. We also assess the performance of our signal detectors using the second dataset. We show that high ratios can be achieved thanks to the lossy compression of the segments without any relevant signal.

## 1. Introduction

Radio Frequency (RF) signals are everywhere. Bluetooth, Wi-Fi, mobile telephony and TV all use RF signals to communicate. The massive adoption of these technologies has pushed forward the development of new standards, thus increasing the number of assigned or licensed frequency bands. By exploiting these bands when they are unused, opportunistic communications aim to use the spectrum in a more efficient way [1,2]. Opportunistic communications are related to cognitive radio, which is a radio that is aware of its internal state and environment and can be dynamically configured to use the best channel. In this sense, it is important to monitor the spectrum for subsequent opportunistic transmissions. This technique is known as spectrum sensing and nowadays is usually performed with a Software Defined Radio (SDR) [3].

Apart from communications, SDRs are also used in other fields such as remote sensing. Earth observation techniques have seen a significant increase in both quality and quantity in recent times, leading to a significant surge in the amount of data produced. Moreover, since remote sensing is usually carried out by satellites with limited storage capacity and bandwidth [4], this growth in data poses significant challenges in terms of data storage and transmission.

Taking into account the extensive presence of RF signals and the constraints of the devices that usually perform sensing tasks, data compression appears to be the key that enables the storage and transfer of this kind of data. For this reason, there are some studies addressing this issue [5,6]. However, they are focused on very specific communications signal types or they do not support lossless (or near-lossless) compression.

This article proposes a new method to compress, either lossless or lossy, generic RF signals. In Section 3, a brief description of FAPEC is provided, i.e., the entropy coder used in our algorithm. In Section 4.1 and Section 4.2, we propose a lossless and a smart lossy pre-processing stage for FAPEC, respectively. Then, in Section 5 and Section 6, we describe the parameters and test files used and the performance of the proposed approach is assessed. Finally, in Section 7, we present our conclusions and state future research lines and possible improvements for the presented methods.

## 2. RF Signals Data Format

In this paper, we deal with the compression of RF data, specifically the signals obtained with a SDR. From communications theory [7], it is known that any pass-band signal s(t) can be expressed as
(1)$$s\left(t\right)=i_{s}\left(t\right)\cdot \cos\left(2\pi f_{0}t\right)-q_{s}\left(t\right)\cdot \sin\left(2\pi f_{0}t\right),$$
where $i_{s}\left(t\right)$ and $q_{s}\left(t\right)$ are the in-phase and quadrature components, respectively, and $f_{0}$ is the carrier frequency.

For simplicity, we may work with the equivalent baseband signal
(2)$$b_{s}\left(t\right)=i_{s}\left(t\right)+jq_{s}\left(t\right).$$
The data to be compressed are the discrete time samples of the signals $i_{s}\left(t\right)$ and $q_{s}\left(t\right)$.

In the dataset used in this work, the samples are 16-bit signed integers and they are interleaved, forming a sequence x(n),n∈{0,⋯,N−1} such that
(3)$$\left\{\begin{array}{c} x\left(2k\right)=i_{s}\left(kT\right)\\ x\left(2k+1\right)=q_{s}\left(kT\right) \end{array}\right.$$
with k∈N and *T* the sampling period.

## 3. The FAPEC Data Compressor

Originally designed for space missions [8], Fully Adaptive Prediction Error Coder (FAPEC) is a highly efficient and extensible data compressor. The advantages of using FAPEC include its resilience in handling noisy or outlier data, its high computing performance and its versatility.

The structure of FAPEC follows a pattern similar to that of other data compressors, with a pre-processing stage followed by an entropy coder [9]. In fact, its name comes from this structure, where the first stage is typically a predictor that estimates the samples and generates a prediction error. The error sequence is then sent to the entropy coder, rather than the original samples. It is worth noting that in some pre-processing stages, such as the one presented in this paper, some side information is also sent to the entropy coder.

FAPEC is equipped with various pre-processing stages, including a basic differential coder, multi-band prediction and a wavelet transform [10]. In its latest version, FAPEC 22.0, an algorithm for water column and bathymetry data, was also added [11,12].

To ensure robustness against data corruption and achieve a high computing performance, FAPEC compresses data in chunks, which typically range from 64 kB to 8 MB. The input sequence is split into several chunks of equal size, and each chunk is processed independently of the others.

## 4. FAPEC Tailoring for RF Data

In Section 2, we described the typical format of RF files: the discrete time samples of the equivalent baseband signal coded as 16-bit signed integers. Now, we shall propose an algorithm for this kind of data. Our approach is based on Linear Predictive Coding (LPC), that is, a linear filter which predicts samples following the model
(4)$$\hat{x}\left(n\right)=\sum_{i=1}^{Q}h_{i}x\left(n-i\right),$$
where $\hat{x}\left(n\right)$ is the predicted sequence, x(n−i) is the previous samples, $h_{i}$ is the filter coefficients and *Q* is the filter order. The prediction error is defined as
(5)$$e\left(n\right)=x\left(n\right)-\hat{x}\left(n\right).$$

There are several reasons that justify using a linear predictor to process RF data. The first one is that a Wide Sense Stationary (WSS) process can always be represented in terms of the optimal linear prediction error. Thus, the filtering described above can also be interpreted as an approximation of x(n) by an AR(Q) process. In addition, we may find papers about IQ data compression that use FLAC to perform the encoding [5]. Finally, linear prediction is a very general and simple method. Taking into account that our target is arbitrary RF signals, a generic method is preferred. Additionally, in the case that the prediction error follows a Gaussian distribution, uncorrelatedness and independence are equivalent, thus linear prediction is optimum [13].

### 4.1. General Aspects of the Proposed Algorithm

In the implementation of a linear predictor as a pre-processing stage for FAPEC, we made some tweaks in order to reduce the computational complexity and improve the performance. In this section, the modifications and the actual implementation are described.

First, it is known that the coefficients $h_{i}$ from Equation (Equation 4) are the solution to the Yule–Walker equations [14]:(6)$$R_{x}h=r_{x}$$
where $R_{x}$ is the autocorrelation matrix with elements $r_{ij}=r_{x}(|i-j|),\;0\leq i,j<Q$, $r_{x}$ the correlation vector with $r_{j}=r_{x}\left(j\right),\;1\leq j\leq Q$ and h the filter coefficients.

Solving the system with Gauss–Jordan elimination has a complexity of $O(Q^{3})$. However, the Toeplitz structure of $R_{x}$ can be exploited and the system can be solved using the Levinson–Durbin recursion [13], with a complexity of $O(Q^{2})$.

Obtaining Equation (Equation 6) involves a statistical approach. However, in this paper, the input sequence x(n) is a finite sequence acquired from an SDR receiver. Consequently, x(n) must be partitioned into subsequences of length N≫Q denoted as $x_{N}\left(n\right)$, where stationarity can be assumed. We should remark that x(n) corresponds to the whole chunk and it is split into shorter sequences of length *N*. The autocorrelation lags are given by the short-term autocorrelation sequence, also known as the autocorrelation method [13]:(7)$$r_{x}\left(i\right)=\sum_{m=0}^{N-1-i}x_{N}\left(m\right)x_{N}\left(m+i\right),\;i\geq 0.$$
It is worth noting that, since the character of the input data is unknown, the parameter *N* is selected by the user when configuring the compressor.

In order to slightly reduce the computational complexity, we may choose not to use all the *N* samples of $x_{N}\left(n\right)$. Instead, we set a training length T≤N and, as it is assumed that the sequence is WSS, coefficients computed with *T* or *N* samples should be very similar. Now, the autocorrelation is calculated as follows:(8)$$r_{x}\left(i\right)=\sum_{m=0}^{T-1-i}x_{N}\left(m\right)x_{N}\left(m+i\right),\;i\geq 0.$$
Using this method, the number of samples used to compute $r_{x}\left(i\right)$ decreases as *i* increases, so i≪T must be used to obtain estimates of good quality. There exists another method to estimate the correlation lags, called the covariance method. This approach uses all *T* samples, so the estimates tend to have better quality. However, the autocorrelation method keeps the Toeplitz structure and the covariance method does not. Further information about the trade-off between these methods can be found in, for instance, reference [15].

Once the autocorrelation was estimated with the autocorrelation method described above, we apply the Levinson–Durbin algorithm to solve the system. In order to improve the performance of the predictor, on each iteration of the recursion, the error is computed, and if it is less than 10% of the initial error or less than 1% of the previous error, the algorithm stops with the filter order used in that iteration. This modification allows the user to select high-filter orders but avoid overfitting and still have a reasonable complexity.

In brief, the proposed algorithm involves splitting the input sequence x(n) into subsequences of length *N* and computing their autocorrelation using T<N samples. Filter coefficients are then obtained via a modified Levinson–Durbin algorithm, which stops when the prediction error reaches a predetermined threshold. Both the coefficients and the prediction errors are subsequently fed to the entropy coder, with the former being quantized for optimal coding. It should be noted that decompression involves calculating $\hat{x}\left(n\right)$ using Equation (Equation 4) and adding it to the prediction error for the corresponding sample. Therefore, it is evident that decompression is less complex than compression.

The algorithm described here can be independently applied to each component. However, the user has the option to enable coupling between components. In this case, the coefficients computed for the first component are reused for the others, resulting in further reduced complexity.

### 4.2. Smart Lossy

The algorithm proposed in the previous section allows to apply a lossy approach (specifically, near-lossless with variable bitrate) by quantizing the input samples just before computing the autocorrelation with Equation (Equation 8). Its implementation rounds the quantized sample instead of truncating it in order to avoid a bias caused by the quantization noise. To avoid error propagation, it is important to apply quantization to the samples rather than the prediction errors.

However, for some use cases such as continuous monitoring in remote sensing, it may be more interesting to adapt the loss level to the features of the data. For instance, if the received signal only contains noise, a high level of losses can be applied. Due to its adaptive capacity, we call this method *smart lossy*.

In order to detect signal features, we propose two different methods: a first one based on spectrum sensing and a second one that takes advantage of the error magnitude computed during the Levinson–Durbin recursion. In practice, the first technique can be used to adjust the parameters of the second, which exhibits a much lower computational load.

The **spectrum sensing method** consists of first estimating the noise power and using this estimate to implement an energy detector. Then, different levels of losses can be applied to what is assumed to be signal or noise, respectively. The problem can be stated as follows:(9)$$\begin{array}{cc} \; & H_{0}:\;x\left(n\right)=w\left(n\right)\\ \; & H_{1}:\;x\left(n\right)=s\left(n\right)+w\left(n\right) \end{array}$$
where $w\left(n\right)∼N\left(0,\sigma_{w}^{2}\right)$ and s(n) is an unknown signal.

Estimating the noise power is a well-known problem, and as such, there exist several techniques that address it [16,17]. Given that we want to be as general as possible regarding the type of signal, we decided to use the Akaike Information Criterion (AIC) [18], as it does not need any knowledge about the signal s(n). The method is as follows: we compute the averaged periodogram of the signal using Welch’s method [19]:(10)$$P_{x}\left(k/L\right)=\frac{1}{K}\sum_{m=0}^{K-1}\underset{\mathrm{Periodogram}}{\underbrace{\frac{1}{L}{\left|\sum_{n=0}^{L-1}x_{m}\left(n\right)e^{\frac{-2\pi j}{L}km}\right|}^{2}}},$$
where *L* is the size of the Fast Fourier Transform (FFT) and *K* is the number of periods of length *N* defined in Section 4.1. For simplicity, we take L=N. In addition, thanks to the Central Limit Theorem (CLT), $P_{x}\left(k/L\right)∼N\left(\sigma_{w}^{2},\frac{\sigma_{w}^{2}}{K}\right)$, and using the AIC, we can find which frequency bins are assumed to represent the signal. The expression of AIC is given by
(11)$$\begin{array}{cc} \; & \mathrm{AIC}(k)=(L-k)\cdot K\cdot \log(\alpha (k))+k(2L-k), \end{array}$$
(12)$$\begin{array}{cc} \; & \alpha \left(k\right)=\frac{\frac{1}{L-k}\sum_{i=k+1}^{L}\lambda_{i}}{\sqrt[{L-k}]{\prod_{i=k+1}^{L}\lambda_{i}}}, \end{array}$$
where $\lambda_{i}=P_{x}\left(i/L\right)$ is the power of the *i*th frequency bin in the averaged periodogram [20].

Knowing the index that minimizes the value of AIC(k)
(13)$$k_{\min}=\underset{k}{\mathrm{argmin}}\;\mathrm{AIC}\left(k\right),$$
the bins $0\leq i<k_{\min}$ are assumed to represent the signal. Thus, an estimate of the noise power can be obtained as
(14)$${\hat{\sigma }}_{w}^{2}=\frac{1}{L-k_{\min}}\sum_{i=k_{\min}}^{L}\lambda_{i}.$$

Once we have an estimate of the noise power, we can proceed with the second step of the method, namely detecting the signal. There are several approaches that deal with this problem, such as the matched filter or cyclostationarity detection [21,22,23,24]. However, these methods need some information about the signal. For this reason, we decided to use an energy detector, as this allows blind detection. The energy detector is given by
(15)$$T\left(x\right)=\sum_{n=0}^{D-1}x{\left(n\right)}^{2}\underset{H_{0}}{\overset{H_{1}}{≷}}\gamma ,$$
where $T\left(x\right)∼\chi_{D,\sigma_{w}^{2}}^{2}$ and γ is the detection threshold.

For a fixed probability of false alarm $P_{F}$, the threshold can be computed with
(16)$$\gamma =Q_{\chi_{D}^{2}}^{-1}\left(P_{F}\right)\cdot {\hat{\sigma }}_{w}^{2},$$
where $Q_{\chi_{D}^{2}}$ is the tail distribution function of the chi-squared distribution with *D* degrees of freedom.

In our approach, the degrees of freedom and the probability of false alarm are user parameters. The former is given by the desired probability of detection and allows to adjust de bias-variance tradeoff, whereas the latter may be used to tune the aggressiveness of the method. For instance, if we want to be cautious, we can set a higher value for $P_{F}$, thus increasing the probability of detecting noise as a signal and performing lossless (or near-lossless) compression more often.

This method has been implemented in the C programming language and has been released under the BSD license. Thus, it can be integrated into FAPEC or other third-party software. The source code is available at [25].

The second method, which we call the **prediction evaluation method**, aims at a simple and fast implementation by reusing quantities already computed in the prediction stage. It relies on the Levinson–Durbin algorithm, from which we take the autocorrelation $r_{x}\left(i\right)$, the training length *T*, the estimated error ϵ, the filter order *Q*, and the LPC coefficients $h_{i}$. From these values, noise power ${\hat{\sigma }}_{w}^{2}$ can be estimated as
(17)$${\hat{\sigma }}_{w}^{2}=\frac{\epsilon }{T},$$
which can be understood as the LPC modeling error. The signal power ${\hat{\sigma }}_{s}^{2}$ can be estimated as
(18)$${\hat{\sigma }}_{s}^{2}=\frac{1}{T}\sum_{i=1}^{Q}h_{i}r_{x}\left(i\right),$$
which can be seen as the LPC modeling success. Thus, for a given sub-sequence of *N* samples where these quantities are determined, we estimate the Signal-to-Noise Ratio (SNR) as
(19)$$\mathrm{SNR}=\frac{{\hat{\sigma }}_{s}^{2}}{{\hat{\sigma }}_{w}^{2}}.$$
Finally, we define the smart lossy quantization step as
(20)$$\delta =\frac{\sqrt{{\hat{\sigma }}_{s}^{2}+{\hat{\sigma }}_{w}^{2}}}{\kappa }$$
where κ is the target dynamic range. That is, $\kappa =2^{\beta }$, where β is the target bits that we want to keep from the digital RF data. Then, depending on the SNR (above or below a given threshold), we can assign different values to κ ($\kappa_{s}$ and $\kappa_{w}$, respectively). This will make lossy compression more or less aggressive (that is, removing more or less bits), setting lower or higher κ values, respectively. Note that δ depends on the total estimated power, meaning that this approach adapts to the actual signal features. Thus, loud signals with a high SNR can still be significantly quantized depending on the κ setting, whereas faint signals also with high SNR may even be left lossless. Note that, in certain types of RF signals such as Global Navigation Satellite System (GNSS), the noise estimation may actually correspond to most of the signal power, whereas the signal estimation may correspond to higher-power parasitic signals. In these cases, the user may configure $\kappa_{w}\gg \kappa_{s}$, meaning that the sub-sequences with high SNR (meaning parasitic signals or interferences) are aggressively compressed, whereas those with low SNR (meaning a “clean” GNSS or spread spectrum signal) can be left with very small losses. Nevertheless, spread spectrum signals exhibit an inherent gain related to the spreading factor, and thus higher loss levels can be used.

Contrary to the near-lossless case (where the quantization is applied to the input samples), in this prediction evaluation method we apply the quantization δ to the prediction errors, which leads to slightly better results in terms of ratio and reconstructed quality. This approach means that we must reconstruct each lossy sample during compression before proceeding to the next sample, meaning some computing overhead but avoiding error propagation.

## 5. Test Setup

The signals used in the following tests come from two different datasets: the first one was obtained with OPS-SAT, a CubeSat by the European Space Agency (ESA) [26]. This consists of seven signals centered at 433.0 MHz, three at 1575.42 MHz and three at 1602.56 MHz. All signals were sampled at 3 MHz. The second one has been captured with a SDRplay RSPdx in Barcelona, Spain, and it consists of three Amplitude Modulation (AM) signals in the medium-wave broadcast band (526.5–1606.5 kHz) and three Automatic Packet Reporting System (APRS) signals [27] in the 2-meter band (144.8 MHz). In this case, the signals were sampled at 15.625 kHz. Both datasets are public and can be downloaded from [28].

To evaluate the lossless data compression performance of our algorithm, we conducted tests in comparison with *gzip*, *Zstd* and FLAC using their default settings. *Gzip* is a commonly used generic compressor that uses a combination of LZ77 [29] and Huffman [30] encoding techniques. *Zstd* is also a general purpose compressor designed to give a compression ratio comparable to that of *gzip*, but much faster. Besides LZ77 and Huffman, it also takes advantage of Finite State Entropy [31]. In the case of FLAC, the format specification sets a maximum sample rate of 655.350 kHz, smaller than the 3 MHz of the OPS-SAT signals. For this reason, we modified the sampling rate in the header to 44.1 kHz. Observe that we have not decimated the signal and that the FLAC performance does not depend on this parameter. Finally, FAPEC has been configured with a period and a training length of N=T=8192 samples and a maximum filter order of Q=16. Channel coupling has also been enabled. In order to show that the value of the training period *T* has a minor impact on the algorithm, we also performed lossless compression tests for T∈{256,512,1024,2048,4096}.

Regarding the smart lossy algorithm, its performance is only assessed with the second dataset. The reason is that we are interested in the quality of the signal after demodulation, and thus, we must know the modulation used, and this is not the case for the first dataset. For the spectrum sensing method, we computed the FFT with N=L=2048 points and computed the detection threshold for a probability of false alarm $P_{F}=0.05$ with D=4096 degrees of freedom. The prediction evaluation method has been configured with an SNR threshold of −5 dB and a target of 0 bits for the noise. The two extreme values for the signal target bits are considered: 16 bits (lossless) and 1 bit ($\kappa_{s}=2$). Observe that this target is not strictly the number of bits used to represent the signal (see Equation (Equation 20) and its explanation).

In order to perform a fair comparison, all tests were forced to operate in single thread mode and were executed on the same Mac mini (M1, 2020) running macOS 13.1.

## 6. Test Results

### 6.1. Lossless and Near-Lossless Compression

This section presents the compression ratio and throughput obtained in the tests. The compression ratio is calculated as the ratio of the size of the original file to the compressed file, while the compression throughput represents the amount of raw data that can be compressed per second.

As can be seen in Figure 1, FAPEC and FLAC exhibit very similar compression ratios. When compared to *gzip* and *Zstd*, FAPEC yields ratios of at least 12% better and 23.4% on average. In this situation, one could consider using a well-known algorithm such as FLAC. However, compression throughput must also be taken into account. In Figure 2, we show that FAPEC is two times faster than FLAC and almost six times faster than *gzip*. Regarding *Zstd*, it is much faster than FAPEC, but as previously shown, it produces lower compression ratios.

In Figure 1, we also show the near-lossless compression ratios for the second and fourth levels of losses. In the first case, the two Least Significant Bits (LSB) are removed and the ratio increases by 28%. In the second case, the three LSB are removed and the ratio increases by 49%. Ratios are remarkably better for GNSS files from the first dataset, given the typically small amplitude of the signals contained therein.

To conclude, the lossless compression results in Figure 3 show that, for a fixed sequence length (N=8192), the value of *T* does not significantly affect the performance of FAPEC. For instance, when T=256, the average compression ratio is 1.765 instead of 1.789, whereas the throughput increases from 142.74 MB/s to 153.02 MB/s. Given this modest decrease in computational efficiency, the rest of the tests are performed with T=N.

### 6.2. Smart Lossy

We conclude the results section by showing the performance of the detector on an APRS signal (D2-APRS-1). In Figure 4, we show that both methods allow the detection of the presence of a signal and the application of different levels of losses. If we set the method to be very aggressive and remove the segments not detected as relevant signals, the compression ratio for this file increases from 2.04 to 32.51 without any errors after demodulation. If we wanted to even further increase the compression ratio, we could also apply lossy compression to the signal segments. For instance, setting $\kappa_{s}=1$ results in a ratio of 104.62, and the signal is still demodulated without any errors.

## 7. Conclusions and Future Work

In this paper, a pre-processing stage for lossless data compression of RF signals has been proposed. In addition, we also proposed two methods to detect the presence of relevant signals, thus allowing the application of different loss levels to noise and signal. In the first method, noise power is estimated with spectrum sensing methods and then an energy detector is implemented using the former estimate of the noise power. On the other hand, the second method relies on the error already computed in each iteration of the Levinson–Durbin algorithm. Hence, the complexity is much lower.

We tested the mentioned algorithms with the FAPEC entropy coder on two different datasets: the first one was obtained with OPS-SAT, a CubeSat by the ESA, and the second one was obtained with a SDRplay RSPdx in Barcelona, Spain. When comparing FAPEC with other compressors such as *Zstd* and *gzip*, we obtain, on average, the best compression ratios on RF signals. On the other hand, the audio coding format FLAC and FAPEC exhibit very similar compression ratios. Regarding the compression throughput, *Zstd* is the fastest algorithm, but the compression ratios are clearly worse. Finally, we showed the performance of the aforementioned detectors using an APRS signal and also how this allows to increase the compression ratios by more than an order of magnitude.

We can outline some future lines of research that could potentially stem from this work. For instance, the proposed lossy algorithm performs variable bitrate encoding, but constant bitrate encoding could also be implemented. Regarding *smart lossy*, other detection methods could be considered. In particular, if it is assumed that noise is Gaussian and the desired signal is not, normality tests could be used as the detector. Observe that we already make this assumption to estimate the noise power. Besides that, more sophisticated algorithms such as neural networks could be employed, at the cost of increasing the complexity and reducing the interpretability. In addition, we intend to perform tests with several GNSS signals and lossy configurations in order to determine the maximum loss levels that would still allow for reliable decoding. Finally, it is known that different modulations require different levels of SNR to be successfully demodulated. For instance, modulations with a large number of symbols require a higher SNR. For this reason, studying the Bit Error Rate (BER) for different modulations and levels of losses could also be insightful.

## Figures and Tables

**Figure 1 sensors-23-03552-f001:**
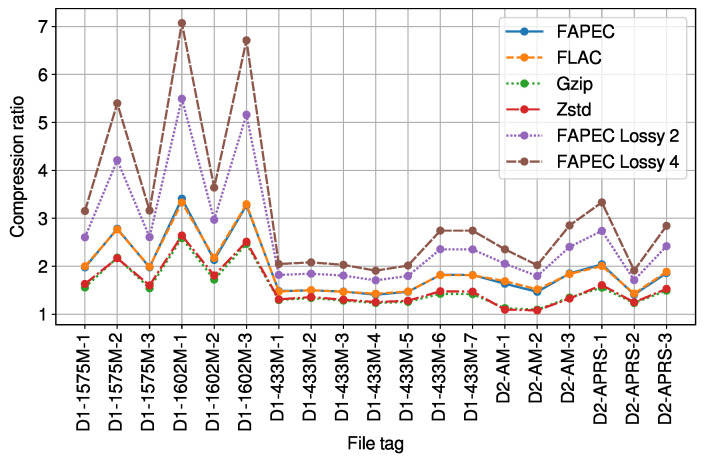
Lossless and near-lossless (levels 2 and 4) compression ratios of the RF signals.

**Figure 2 sensors-23-03552-f002:**
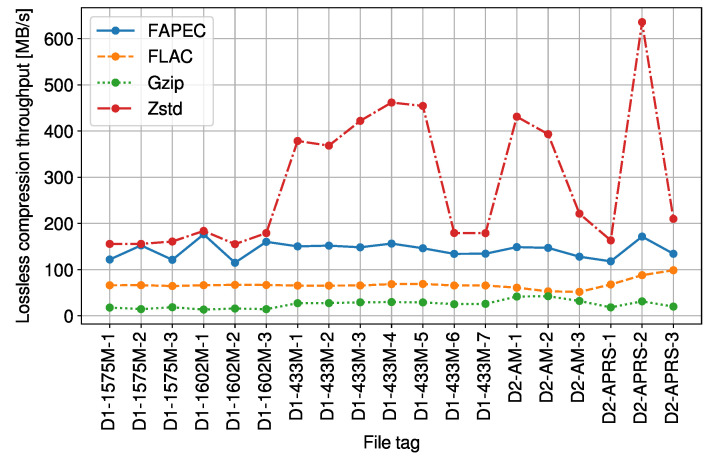
Lossless compression throughput of the RF signals.

**Figure 3 sensors-23-03552-f003:**
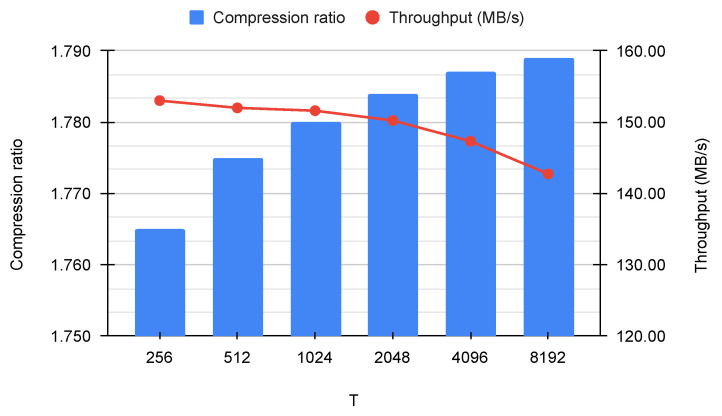
Average compression ratio and throughput (complete dataset) for N=8192 and different values of *T*.

**Figure 4 sensors-23-03552-f004:**
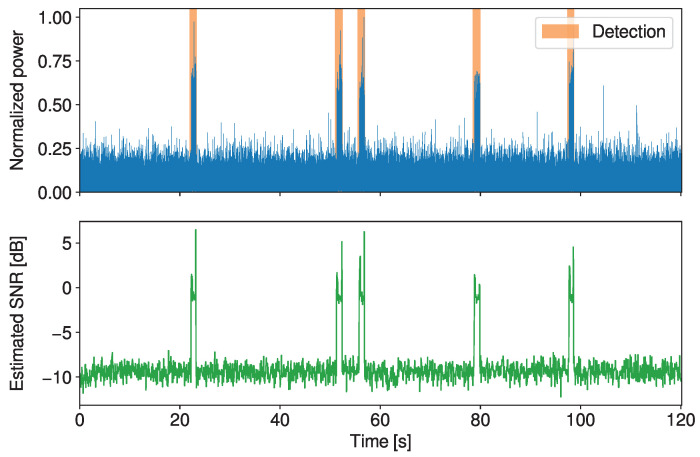
Normalized power, detected bands and estimated SNR of an APRS signal (D2-APRS-1).

## Data Availability

The data presented in this study are openly available in *Dataset from: Context-Aware Lossless and Lossy Compression of Radio Frequency Signals* at 10.21227/ccdy-s283, reference number [28]. The FAPEC data compressor is available at www.dapcom.es (accessed on 20 February 2023).

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
