# Peer review of "Context-Aware Lossless and Lossy Compression of Radio Frequency Signals"

_sensors, 2023, doi:10.3390/s23073552_

Round 1

Reviewer 1 Report

The compression is carried out by integrating various available methods on various data sets.  It looks to get results by hit and trial integration of compression techniques.  English may be improved for this high-quality journal.  "I", "we" is generally not used in the text. Levinson-Durbin and Fully Adaptive Prediction Error Coder have been utilized efficiently to achieve compression.  Compression and error detection both are carried out using different RF data samples. 

After English text and abstract modification paper may be accepted for publication.  In abstract it is mentioned " Good Compression" while it should be quantitative value or number and not generic.

Reviewer 2 Report

See the attachment. 
